# Oleate Hydratase in *Lactobacillus delbrueckii* subsp. *bulgaricus* LBP UFSC 2230 Catalyzes the Reversible Conversion between Linoleic Acid and Ricinoleic Acid

Gabriela Christina Kuhl,[a,b,f] Ricardo Ruiz Mazzon,[b] Brenda Lee Simas Porto,[c] Tâmela Zamboni Madaloz,[d] Guilherme Razzera,[d] Daniel De Oliveira Patricio,[b] Kevin Linehan,[e,f,g] Grace Ahern,[e,f,g] Harsh Mathur,[f] Paul Ross,[e] Catherine Stanton,[e,f] Juliano De Dea Lindner[a]

[a]Department of Food Science and Technology, Federal University of Santa Catarina (UFSC), Florianópolis, Brazil
[b]Department of Microbiology, Immunology, and Parasitology, Federal University of Santa Catarina (UFSC), Florianópolis, Brazil
[c]Department of Chemistry, Institute of Exact Sciences, Federal University of Minas Gerais, Belo Horizonte, Minas Gerais, Brazil
[d]Department of Biochemistry, Federal University of Santa Catarina (UFSC), Florianópolis, Brazil
[e]APC Microbiome Ireland, University College Cork (UCC), Cork, Ireland
[f]Teagasc Food Research Centre, Fermoy, Ireland
[g]School of Microbiology, University College Cork (UCC), Cork, Ireland

**ABSTRACT** Conjugated linoleic acid (CLA) has been the subject of numerous studies in recent decades because of its associated health benefits. CLA is an intermediate product of the biohydrogenation pathway of linoleic acid (LA) in bacteria. Several bacterial species capable of efficiently converting LA into CLA have been widely reported in the literature, among them *Lactobacillus delbrueckii* subsp. *bulgaricus* LBP UFSC 2230. Over the last few years, a multicomponent enzymatic system consisting of three enzymes involved in the biohydrogenation process of LA has been proposed. Sequencing the genome of *L. delbrueckii* subsp. *bulgaricus* LBP UFSC 2230 revealed only one gene capable of encoding an oleate hydratase (OleH), unlike the presence of multiple genes typically found in similar strains. This study investigated the biological effect of the OleH enzyme of *L. delbrueckii* subsp. *bulgaricus* LBP UFSC 2230 on the hydration of LA and dehydration of ricinoleic acid (RA) and its possible role in the production of CLA. The OleH was cloned, expressed, purified, and characterized. Fatty acid measurements were made by an internal standard method using a gas chromatography-coupled flame ionization detector (GC-FID) system. It was found that the enzyme is a hydratase/dehydratase, leading to a reversible transformation between LA and RA. In addition, the results showed that *L. delbrueckii* subsp. *bulgaricus* LBP UFSC 2230 OleH protein plays a role in stress tolerance in *Escherichia coli*. In conclusion, the OleH of *L. delbrueckii* subsp. *bulgaricus* LBP UFSC 2230 catalyzes the initial stage of saturation metabolism of LA, although it has not converted the substrates directly into CLA.

**IMPORTANCE** This study provides insight into the enzymatic mechanism of CLA synthesis in *L. delbrueckii* subsp. *bulgaricus* and broadens our understanding of the bioconversion of LA and RA by OleH. The impact of OleH on the production of the *c*9, *t*11 CLA isomer and stress tolerance by *E. coli* has been assisted. The results provide an understanding of the factors which influence OleH activity. *L. delbrueckii* subsp. *bulgaricus* LBP UFSC 2230 OleH presented two putative fatty acid-binding sites. Recombinant OleH catalyzed both LA hydration and RA dehydration. OleH was shown to play a role in bacterial growth performance in the presence of LA.

**KEYWORDS** biohydrogenation, heterologous expression, homology modeling, *cis*-9, *trans*-11 CLA

Address correspondence to Juliano De Dea Lindner, juliano.lindner@ufsc.br.

Lactic acid bacteria (LAB) have extensively been used as microbial starter cultures for the fermentation of dairy products to promote health (1). These microorganisms act as cellular factories for the synthesis of secondary metabolites with functional properties, such as the production of conjugated linoleic acid (CLA) (2–5). CLA refers to a collective term describing a group of positional and geometric isomers of linoleic acid ($c9$, $c12$-$C_{18:2}$; LA) with conjugated double bonds (6). There are 28 possible CLA isomers (7), of which the predominant form is $c9$, $t11$-$C_{18:2}$ (8). The $c9$, $t11$-$C_{18:2}$ isomer has been shown to exert biological effects on human metabolism, such as anticarcinogenic, antidiabetic, antiatherosclerotic, antiosteoporosis, and immune system stimulation (9, 10).

Predominantly found in dairy products from ruminants (11), CLA isomers are formed as intermediates during LA biohydrogenation to stearic acid (12). The biohydrogenation of LA into CLA isomers has been reported as a multicomponent enzymatic system encoded in the *Lactobacillus* genome (13). In our previous work, genome sequencing of *Lactobacillus delbrueckii* subsp. *bulgaricus* LBP UFSC 2230 (BioProject accession number PRJNA615231; SRA accession number SRR11741240) confirmed the presence of a single gene (*oleH*) capable of expressing an enzyme which was identified as an oleate hydratase and not the presence of other genes which are typically found in other such lactobacilli strains, generally involved in this multienzyme process (14).

In a previous study, several LAB strains were screened for their ability to produce CLA from LA, and we found *L. delbrueckii* subsp. *bulgaricus* LBP UFSC 2230 as a potential strain to synthesize the $c9$, $t11$-$C_{18:2}$ isomer (3). The current study aimed to characterize the oleate hydratase (EC 4.2.1.53) enzyme from *L. delbrueckii* subsp. *bulgaricus* LBP UFSC 2230 by analyzing the enzymatic activity (*in vivo* and *in vitro*) of the expressed protein in a heterologous host.

## RESULTS

**Growth rate measurements.** The purpose of this assay was to establish the culture conditions for maximum growth rate and detect LA concentration that influences the growth of *L. delbrueckii* subsp. *bulgaricus* LBP UFSC 2230. According to the maximum growth rate ($\mu_{max}$), LA appeared not to be sufficient to completely inhibit bacterial growth at all tested concentrations, even though the difference in growth rate between the maximum concentration (2.0 mg/ml LA) and the control was statistically significant (analysis of variance [ANOVA], $P < 0,0001$).

**Cloning, expression, and purification of *L. delbrueckii* subsp. *bulgaricus* LBP UFSC 2230 putative OleH protein.** The putative OleH ORF was cloned into pET28a (Novagen, Darmstadt, Germany) and expressed in *E. coli* BL21(DE3). The complete 1,776-bp nucleotide sequence (HCO48_RS09600) from *L. delbrueckii* subsp. *bulgaricus* LBP UFSC 2230 encoded a protein of 591 amino acids (MBN6090876.1). The amino acid sequence database comparison revealed that the putative OleH *L. delbrueckii* subsp. *bulgaricus* protein was 100% identical to the homologous sequence from *L. delbrueckii* subsp. *bulgaricus* KW14_3 (GenPept accession no. WP_129335718.1). The soluble fraction of crude extract was eluted from HisTrap FF by a linear gradient of imidazole. The purified protein displayed a single band of approximately 67 kDa corresponding to the expected molecular weight of the putative OleH protein. Immunoblotting tracing the His-tagged protein with anti-His tag antibodies confirmed the presence of soluble protein. The purified OleH was used for protein sequencing.

**Protein sequencing.** Matrix-assisted laser desorption ionization–time of flight mass spectrometry (MALDI-TOF MS) was performed for protein identification. According to the NCBIprot database, the recombinant protein was identified as oleate hydratase from *Lactobacillus delbrueckii*, with a molecular mass of 67.560 kDa, isoelectric point (pI) of 5.25, sequence coverage of 34% (the ratio of portion sequence covered by matched peptide to the full length of the protein sequence), PMF of 17 (the number of matched peptides resulting from peptide mass fingerprinting), score of 201 [score is $-10 \times \log(P)$, where $P$ is the probability that an observed match is a random event], and $P$ value of <0.05.

**TABLE 1** Estimated $\Delta G$ and residue interactions from molecular docking analysis in OleH three-dimensional homology model

| Ligand | Cavity name | $\Delta G$ (kcal/mol) | Polar contact(s) | Nonpolar contacts |
|---|---|---|---|---|
| LA control (PDB ID 4IA6)[a] | A | $-5.3 \pm 0.2$[b] | | I[149], I[153], M[154], I[205], M[547], M[550] |
| FAD control (PDB ID 2B9W)[a] | B | $-11.7 \pm 0.2$[b] | A[17], K[45], M[62], F[408] | I[12], G[13], A[14], G[15], P[16], L[36], E[37], R[38], T[39], G[44], M[58], G[59], A[60], Y[67], V[254], Y[281], V[283], W[368], Y[370], Y[382], G[400], E[401], G[406], N[407], V[411] |
| LA[c] | A | $-5.2 \pm 0.2$ | | I[149], V[150], I[153], A[201], I[205], I[208], L[543], T[547], L[550], I[556], V[568], A[571], M[572] |
| | B | $-7.2 \pm 0.3$ | Y[411] | G[80], R[81], E[82], T[184], M[185], A[187], F[219], W[343], I[378], H[393], W[409], F[507] |
| RA[c] | A | $-5.4 \pm 0.4$ | | I[149], V[150], K[151], I[153], M[154], A[201], I[205], I[208], A[546], L[550], L[561], P[564] |
| | B | $-7.3 \pm 0.4$ | Y[411] | G[80], R[81], E[82], T[184], M[185], L[217], F[219], W[343], G[377], I[378], T[391], H[393], L[413] |
| FAD[c] | B | $-11.5 \pm 0.4$ | L[32], Q[224], T[288], I[292], S[314], N[496], T[508] | G[31], A[33], E[57], G[79], I[76], I[77], G[290], S[291], V[293], Y[471], G[495], T[509], S[512] |

[a]The presented interaction contacts were mapped from crystal structures 4IA6 and 2B9W using LigPlot (43).
[b]Affinity values were estimated from ligand redocking using the crystal structures 4IA6 and 2B9W. $\Delta G$ values are expressed as the mean $\pm$ SD.
[c]LA, linoleic acid; FAD, flavin adenine dinucleotide; RA, ricinoleic acid.

**Homology modeling and molecular docking of *L. delbrueckii* subsp. *bulgaricus* oleate hydratase.** To evaluate ligand-binding propensies in OleH from *L. delbrueckii* subsp. *bulgaricus* LBP UFSC 2230, a three-dimensional homology model was built using the best PDB hit as a template (PDB ID 4IA6 from *Lactobacillus acidophilus*) with 73% sequence identity. A superimposition of the OleH from the *L. delbrueckii* subsp. *bulgaricus* model with the template 4IA6 showed a high-quality model with a backbone root mean square deviation of 0.29 Å. The three-dimensional model was subjected to structural quality assessment and validated for *psi* and *phi* torsion positions using the Ramachandran plot. The analysis revealed that 90.3% of the residues of the built model were placed in the most favored regions, indicating a high quality of the predicted structure. The structural model was used for ligand-binding prediction through molecular docking analysis.

The interaction of OleH from *L. delbrueckii* subsp. *bulgaricus* with fatty acids (LA and ricinoleic acid [RA]) and flavin adenine dinucleotide (FAD) was evaluated by $\Delta G$ estimation (free energy data) (Table 1). Two cavities were mapped in the OleH three-dimensional protein model (named cavities A and B) (Fig. 1A). Fatty acid-binding sites were identified in the two cavities, and the FAD-binding site was docked only in cavity B. When the fatty acid-binding sites were analyzed, the overall affinity in cavity B (Table 1; Fig. 1C and F) presented lower $\Delta G$ values ($-6.9$ kcal/mol) than in cavity A (around $-5$ kcal/mol). The lower $\Delta G$ values were related to the polar interactions between the Y[411] hydrogen donor and the fatty acid polar head groups (Fig. 1C and F). Besides the lower $\Delta G$, with no polar contacts for fatty acid binding in cavity A, the values obtained were similar to the LA control in the 4IA6 crystal structure (Table 1). When LA and RA were compared, no relevant differences in $\Delta G$ were observed. The mapped OleH from *L. delbrueckii* subsp. *bulgaricus* FAD-binding site presented similar $\Delta G$ to the FAD control crystal structure 2B9W (around $-11$ kcal/mol). The binding site involved polar interactions with the residues T[288] and S[314] (related to sugar ring), L[32] involved hydrogen bound with flavin at isoalloxazine oxygen, I[292] and Q[224] were associated with the FAD phosphate moiety, and T[508] and N[496] hydrogen bound to the bridge between flavin and phosphates.

**LA tolerance of OleH-expressing *E. coli*.** This assay was conducted to evaluate the growth kinetics of *E. coli* expressing recombinant OleH in an LA-rich medium. To determine if *oleH*-expressing *E. coli* BL21(DE3) improved LA tolerance, the ability of the *E. coli* strain to grow in the presence of different concentrations of LA was investigated. The *E. coli* BL21(DE3) culture not induced by isopropyl-$\beta$-D-1-thiogalactopyranoside (IPTG) served as negative control. Cultures without LA supplementation served as positive controls.

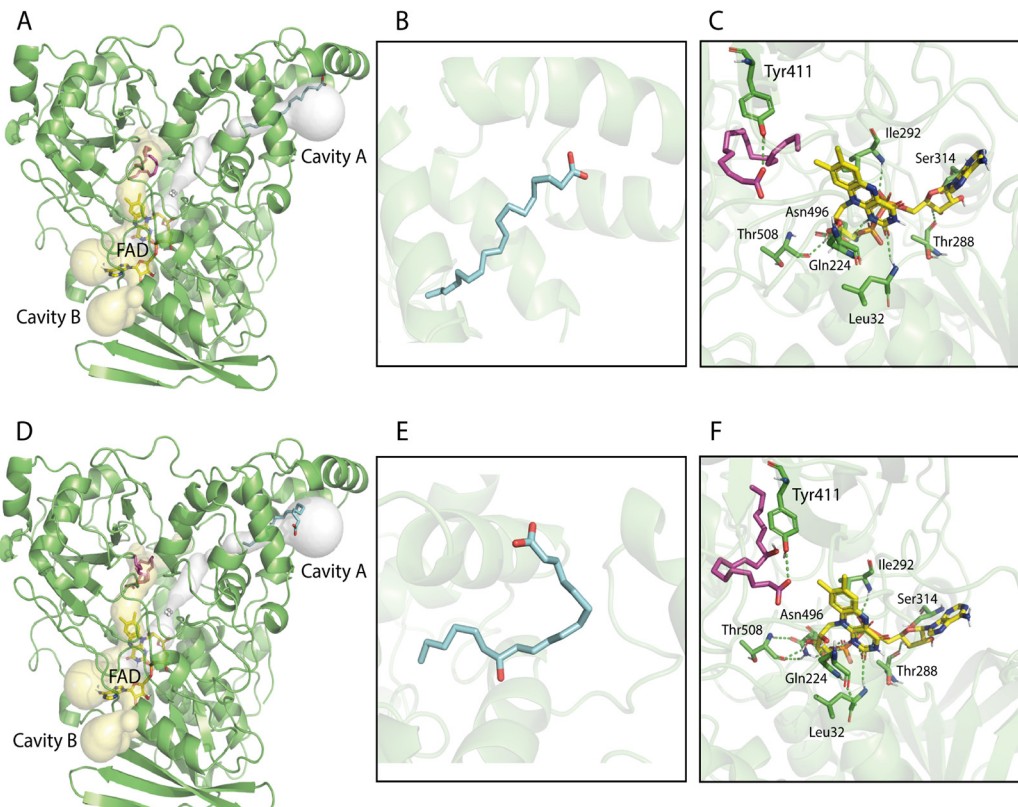

**FIG 1** Oleate hydratase *Lactobacillus delbrueckii* subsp. *bulgaricus* LBP UFSC 2230 three-dimensional homology modeling and molecular docking. (A to C) Linoleic acid (LA) binding. (D to F) Ricinoleic acid (RA) binding. (A and D) The overall three-dimensional model shows the oleate hydratase fold. The cavities A and B are represented by internal tunnel access mapped with Caver software (38) in white and yellow, respectively. (B) Closeup view of LA bound to cavity A ($\Delta G$, $-5.1$ kcal/mol). (C) Closeup view of the LA (magenta) and flavin adenine dinucleotide (FAD; yellow) bound to cavity B ($\Delta G$, $-6.9$ and $-11.1$ kcal/mol, respectively). Polar contact residues are represented in green and hydrogen bonds by dashed lines.

To assess the LA tolerance of *E. coli* BL21(DE3), the ability of cultures to grow was evaluated in LB broth supplemented with 0.1, 0.5, and 1.0 mg/ml LA. During this preliminary experiment, in the absence of the inducer IPTG, it was found that at 0.1 mg/ml LA, growth showed no significant difference from the positive control. However, at 0.5 and 1.0 mg/ml LA, growth was significantly different ($P < 0.0001$) from the positive control, indicating that the inhibitory concentration starts at a concentration of 0.5 mg/ml LA.

In the presence of 0.5 mg/ml LA, IPTG-induced *E. coli* BL21(DE3) culture increased exponentially with time. The growth variation between the IPTG-treated culture and the positive control was not significantly different. Compared to the negative control (in the absence of the inducer IPTG), both IPTG-treated and the positive-control cultures showed significant differences ($P < 0.0001$) (Fig. 2A). Although IPTG-treated culture was significantly ($P < 0.0001$) tolerant to LA at 1.0 mg/ml concentration, the cell growth appears to have been partially inhibited. The IPTG-treated culture displayed substantially increased growth compared to the non-IPTG-induced culture (negative control). However, compared to the positive control, the growth variation also showed a significant difference ($P < 0.0001$) (Fig. 2B).

In this assay, the experimental results showed that the OleH effect on bacterial growth decreased with the increasing LA concentration in the media (Fig. 3).

**Oleate hydratase activity assay.** It has previously been reported that *L. delbrueckii* subsp. *bulgaricus* LBP UFSC 2230 was the most efficient CLA producer among 13 strains of the same species (3). In the current study, we assessed the ability of OleH from *L.*

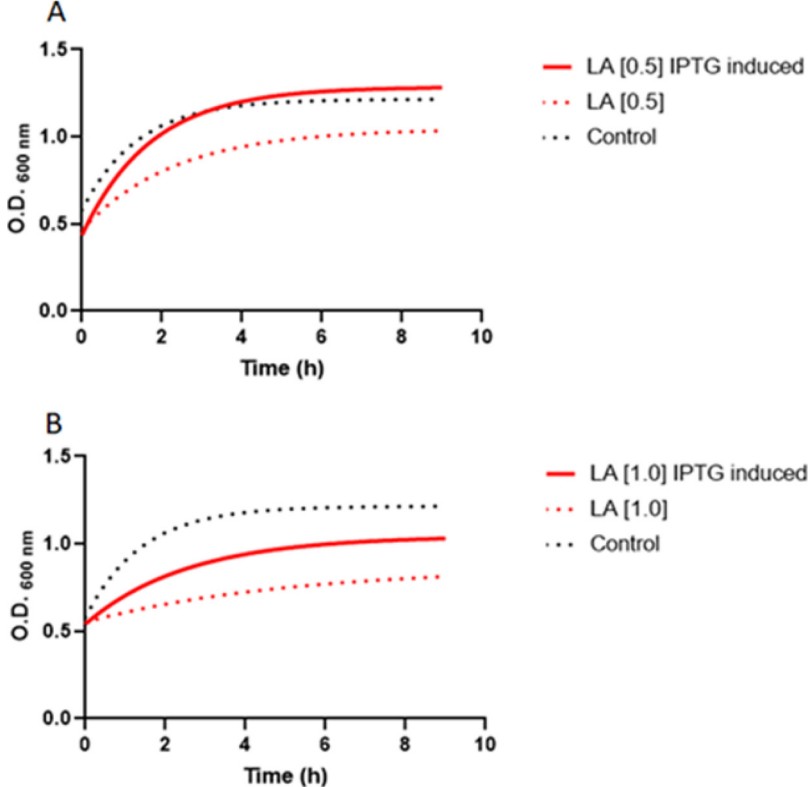

**FIG 2** Linoleic acid (LA) growth inhibition. (A) LA concentration 0.5 mg/ml; (B) LA concentration 1.0 mg/ml. IPTG, isopropyl-$\beta$-D-1-thiogalactopyranoside; control, absence of LA.

*delbrueckii* subsp. *bulgaricus* LBP UFSC 2230 to produce CLA. The enzymatic activity of OleH from *L. delbrueckii* subsp. *bulgaricus* was examined using LA or RA as the substrates. The enzyme was found to be more stable at pH 6.0 at a temperature of 37°C. Fully washed *E. coli* cells induced by IPTG, their cell lysates, and the purified enzyme were incubated in individual separate reaction mixtures containing each fatty acid. The products were analyzed by GC.

None of the investigated substrates were converted into CLA ($c9$, $t11$-C$_{18:2}$) at a detectable level (Table 2). However, the recombinant enzyme has been used successfully for the production of hydroxylated fatty acid from LA. LA was catalyzed by hydration of the *cis* double bond at the Δ12 position, leading to the formation of 12-hydroxy-9-*cis*-octadecenoic acid. The reaction time did not influence the RA formation ($P > 0.9999$;

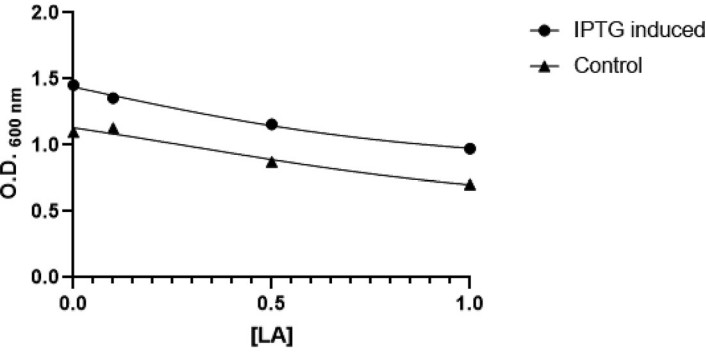

**FIG 3** *Escherichia coli* BL21(DE3) overexpressing OleH from *Lactobacillus delbrueckii* subsp. *bulgaricus* LBP UFSC 2230 growth at different linoleic acid (LA) concentrations (mg/ml). IPTG, isopropyl $\beta$-D-1-thiogalactopyranoside; control, culture not induced with IPTG.

**TABLE 2** Products of enzymatic reaction of OleH from *Lactobacillus delbrueckii* subsp. *bulgaricus* LBP UFSC 2230

| Substrate[d] | Product ($\mu$g/g)[c] | |
| --- | --- | --- |
| | RA | LA |
| LA[a] (15 min) | 11.03 | |
| LA[a] (30 min) | 32.31 | |
| LA[a] (45 min) | 21.08 | |
| LA[a] (60 min) | 33.57 | |
| LA[b] (60 min) | 32.24 | |
| RA[a] (60 min) | | 4.23 |
| RA[b] (60 min) | | 3.70 |
| Retention time | 37.93 | 39.51 |

[a]Purified recombinant protein reaction.
[b]Biocatalysis reaction.
[c]Data represent $\mu$g/g total fatty acids.
[d]RA, ricinoleic acid; LA, linoleic acid.

ANOVA) (Fig. 4). Furthermore, the same product was also obtained through whole-cell biocatalysis.

The reaction of purified OleH with RA led to the formation of the corresponding fatty acid with double-bond *cis* at position $\Delta12$, suggesting that the *oleH* gene product acts as a hydratase that is also responsible for the reversible reaction between LA and RA. During the enzymatic reaction of purified OleH from *L. delbrueckii* subsp. *bulgaricus*, 25% LA was formed, and during the biocatalysis of OleH from *L. delbrueckii* subsp. *bulgaricus* expressed by whole cells of *E. coli* BL21(DE3), the formation of 23% of LA occurred.

## DISCUSSION

Researchers have proposed a hypothesis of a multicomponent enzymatic system consisting of three enzymes involving the biohydrogenation process of LA (15). However, the genome sequencing of *L. delbrueckii* subsp. *bulgaricus* LBP UFSC 2230 suggested that only one gene capable of encoding an oleate hydratase (*oleH*) enzyme (14) based on homology testing. In a previous study, our group identified that the production of the CLA isomers in a coculture of *L. delbrueckii* subsp. *bulgaricus* LBP UFSC 2230 and *Streptococcus thermophilus* strain 360 is conditioned to the predominant bacterial species in the starter culture. Data indicated that when the proportion of

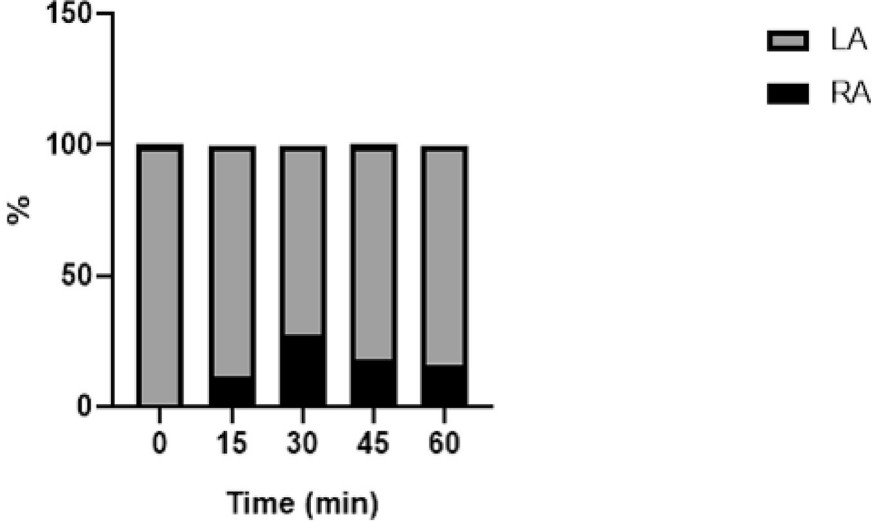

**FIG 4** Time course of ricinoleic acid (RA) formation from linoleic acid (LA) during the OleH-mediated enzymatic reaction from *Lactobacillus delbrueckii* subsp. *Bulgaricus* LBP UFSC 2230.

*L. delbrueckii* subsp. *bulgaricus* LBP UFSC 2230 exceeded that of *S. thermophilus* strain 360, the LBP UFSC 2230 culture demonstrated the potential ability to synthesize the $c9$, $t11$-$C_{18:2}$ isomer, but the production of the $t10$, $c12$-$C_{18:2}$ was not observed. Nevertheless, in the opposite condition, there was a significant detection of the latter isomer (3).

According to the metabolic pathway suggested by Rodríguez-Alcalá et al. (16), the production of the $t10$, $c12$-$C_{18:2}$ isomer is attributed to the linoleate isomerase enzyme, which corroborates with the genomic data presented above, suggesting that the *L. delbrueckii* subsp. *bulgaricus* LBP UFSC 2230 of the coculture was responsible for the production of $c9$, $t11$-$C_{18:2}$ CLA isomer through an alternative metabolic route. Hydratase activity has been associated with the first step of LA biohydrogenation, leading to the formation of hydroxy fatty acids as intermediates in the CLA pathway (16, 17). In addition, a possible pathway for CLA synthesis from RA by direct transformation to CLA isomers has been reported in the past years (18, 19).

These data prompted us to speculate the role of OleH from *L. delbrueckii* subsp. *bulgaricus* LBP UFSC 2230 in CLA production as a possible two-step process in which RA is produced followed by CLA isomer synthesis by the same enzyme. However, it is evident in this study that the OleH from *L. delbrueckii* subsp. *bulgaricus* does not directly participate in the conversion of LA into CLA and that this bacterium follows a different biochemical pathway to produce CLA.

Instead, recombinant OleH catalyzed the conversion of RA to LA, indicating that OleH from *L. delbrueckii* subsp. *bulgaricus* LBP UFSC 2230 can perform the reversible conversion of LA to RA, as verified by Gao et al. (20). The authors elucidated that bifidobacterial myosin cross-reactive antigen (MCRA; fatty acid hydratase) could transform 90-hydroxy-*cis*-12-octadecenoic acid (10-HOE) into LA reversibly, and then LA was converted into CLA through LA isomerase. Since annotation of any putative linoleate isomerase in the *L. delbrueckii* subsp. *bulgaricus* LBP UFSC 2230 genome (14) was not evident, more studies will be carried out in the future to fully unravel the complex mechanisms and pathways involved in CLA production in this strain.

While some oleate hydratase enzymes are selective for double bonds at the Δ9 position, OleH from *Streptococcus pyogenes*, for example, catalyzes the hydration of *cis* double bonds at both the Δ9 and Δ12 positions (21). In this study, OleH from *L. delbrueckii* subsp. *bulgaricus* LBP UFSC 2230 appears to be selective for hydration of *cis* double bonds in the Δ12 position. Also, OleH from *Elizabethkingia meningoseptica* catalyzes the reversible hydration of the *cis* double bond of oleic acid (*cis*-9-18:1) to 10 (R)-hydroxy-18:0, which, due to reverse dehydration, can also be converted to *trans*-10-8:1 or *cis* 9-18:1 (22, 23), corroborating our reverse reaction data.

Although, here, the production of CLA isomers was not observed, the enzymatic reactions that we identified can be useful in modulating the properties of fatty acids in foods. As an example, the dehydration reaction catalyzed by OleH from *L. delbrueckii* subsp. *bulgaricus* LBP UFSC 2230 could increase *cis* dehydration, reducing the amounts of trans-fatty acids in foods.

Based on three-dimensional homology modeling and molecular docking analysis, the *L. delbrueckii* subsp. *bulgaricus* LBP UFSC 2230 OleH presented two putative fatty acid-binding sites. The structure used as a model template (PDB ID 4IA6) from *L. acidophilus* showed linoleic acid bound in cavity A. This region has been considered a portal entrance for fatty acids (23). The tunnel access analysis suggests the connection between cavities A and B, where the fatty acid may reach the FAD-binding site. The *in silico* data showed fatty acid polar interaction with $Y^{414}$ and a higher affinity to the binding site at cavity B, indicating the movement from cavity A to B. In the 4IA6 structure, FAD was not crystallized, probably due to high flexibility in this region. The proposed FAD-binding site involved the conserved signature $GXGXXG(X)_{17/23}E$ (23). The FAD-binding site mapped by docking analysis in *L. delbrueckii* subsp. *bulgaricus* LBP UFSC 2230 OleH showed nonpolar interactions with $G^{31}$, $A^{33}$, $E^{57}$ suggesting that the conserved motif $G^{29}G^{30}G^{31}L^{32}A^{33}G^{34}(X)_{23}E^{57}$ is related to FAD binding.

With regard to LA tolerance, LA biohydrogenation pathways have been proposed to function as a detoxifying mechanism for bacteria (24). The tolerance level to free polyunsaturated fatty acids is different depending on the microorganism and its growth conditions (25). Concerning LA tolerance by *Lactobacillus* spp., some authors have reported different bacterial growth inhibitory concentrations, ranging from 100 $\mu$g/ml to 5 mg/ml (26, 27). A screening to identify higher LA-tolerant bacterial strains has been suggested as a parameter to consider CLA synthesis (28). In this study, recombinant OleH appeared to play a role in bacterial growth performance in the presence of LA by transforming it into a less toxic fatty acid. Our data suggested that the presence of OleH in the cell increased LA tolerance by the culture at concentration of 0.5 mg/ml LA, at least when this expression system was used.

Since the induction of the target protein expression was useful to maintain tolerance at this concentration, the partial inhibition at 1.0 mg/ml may not be attributable to the occasional toxicity of target protein to the cell (29). In this case, the reduced growth rate appears to be associated with the LA dose-response inhibition. The nature of LA inhibitory activity remains unknown. Nevertheless, some authors have associated it with an increase in the permeability of the cell membrane of bacteria (30, 31). Volkov et al. (21) hypothesized that the hydration of unsaturated fatty acids could provide a detoxification mechanism of unsaturated fatty acids' detrimental effect on the cytoplasmic membrane.

Many bacteria were found to detoxify unsaturated fatty acids by the MCRA protein, converting them into hydroxy fatty acids. According to Kishino et al. (18), CLA-HY, which belongs to the MCRA family, is responsible for detoxifying unsaturated fatty acids in *Lactobacillus plantarum*. In fact, due to the immunomodulatory activity of hydroxy fatty acids, Bergamo et al. (32) have considered these molecules similar to CLA (*c*9, *t*11-C$_{18:2}$) as functional components in food and pharmaceutical products.

In this study, we describe the enzymatic and physicochemical characteristics of OleH from *L. delbrueckii* subsp. *bulgaricus* LBP UFSC 2230, which catalyzes an early stage of saturation metabolism of LA. It was found that the enzyme is a reversible hydratase/dehydratase, showing activity in the presence of FAD. In conclusion, OleH from *L. delbrueckii* subsp. *bulgaricus* LBP UFSC 2230 could catalyze the hydration and dehydration of the tested substrates, leading to the reversible transformation between LA and RA. Despite this finding, no CLA was generated independently of the substrate tested. In addition, the enzyme showed detoxification activity as a physiological adaptation to environments rich in unsaturated fatty acids. Overall, our experimental evidence reported herein demonstrates that the OleH protein encoded by *L. delbrueckii* subsp. *bulgaricus* LBP UFSC 2230 plays a role in stress tolerance and catalyzes the initial stage of saturation metabolism of LA, although it has not converted the substrates directly into CLA.

## MATERIALS AND METHODS

***Lactobacillus delbrueckii* subsp. *bulgaricus* LBP UFSC 2230 strain and growth rate measurements.** *L. delbrueckii* subsp. *bulgaricus* strain LBP UFSC 2230, isolated from Italian Grana Padano cheese, has been previously demonstrated to be the most efficient CLA producer among 13 strains of the same species tested (3). Reaction conditions included incubation of the inoculum in de Man-Rogosa-Sharpe (MRS) broth medium (Merck, Darmstadt, Germany) at 37°C for 18 h under anaerobic conditions.

The growth rate of *L. delbrueckii* subsp. *bulgaricus* LBP UFSC 2230 was measured as the maximum increase in optical density (OD) over time during exponential growth. Bacterial growth was assessed by absorbance at 600 nm wavelength. Incubation and measurements were performed in a multiplate reader (SpectraMax Paradigm Molecular Devices, Sunnyvale, CA, USA) at 37°C for 12 h, with data collection every hour. To detect sensitivity of the strain to LA, overnight-activated strains were inoculated in MRS broth with concentrations of 0.1, 0.5, 1.0, and 2.0 mg/ml LA. Cultures without LA supplementation served as negative controls. Microbial growth curve was expressed as the Napierian logarithm of the microbial concentration (OD) against time (h). In this curve, the parameter-specific growth rate ($\mu_{max}$) was defined as the slope of the tangent line at the inflection point. The parameter lag time ($\lambda$) was defined as the intercept of this tangent line with the value of the initial microbial density according to Zwietering et al. (33). To assess multiple comparisons between two continuous variables, a multiple linear regression was fitted (lm and analysis of variance [ANOVA] functions in R) for continuous response

**TABLE 3** Primers used in polymerase chain reactions

| Primer | Restriction site | Sequence (5'–3') |
|---|---|---|
| OLEH_fw | NdeI | TTT<u>CATATG</u>TATTATTCAAACGGTAATTACGAAGC |
| OLEH_rv | SalI | TTT<u>GTCGAC</u>TGACCAAAAGAAAAAGAGAAGCAATTTGC |

data. The curve fitting and analysis were performed using GraphPad Prism Software, version 7.00 (GraphPad Software Inc., CA, USA).

**DNA manipulation and plasmids constructions.** Genomic DNA was isolated from *L. delbrueckii* subsp. *bulgaricus* LBP UFSC 2230 using the GenElute bacterial genomic DNA kit (Sigma-Aldrich, Dublin, Ireland). Two primers were designed based on the genomic sequence of *L. delbrueckii* subsp. *bulgaricus* LBP UFSC 2230 (Table 3) to amplify the *oleH* gene encoding the putative oleate hydratase enzyme (OleH). For the heterologous expression assay, the *oleH* gene was amplified by PCR with the primers OLEH_fw/OLEH_rv.

PCRs were performed with Phusion High-Fidelity DNA polymerase (NEB, Ipswich, MA, USA) as described by the supplier. The underlined bases in these sequences indicate the restriction enzyme recognition sites incorporated into each primer. The amplified products were subjected to a double-digest with the corresponding restriction enzymes, followed by an overnight ligation reaction at 16°C with T4 DNA ligase (Roche, Dublin, Ireland) into the expression vector pET-28a (Novagen, Darmstadt, Germany), resulting in the construct pOLEH.

The recombinant plasmid was transformed into chemically competent *Escherichia coli* using One Shot TOP10 (Invitrogen, Carlsbad, USA). Positive cells were screened on Luria-Bertani (LB) agar containing kanamycin (50 $\mu$g/ml) to select pOLEH. The GenElute Plasmid miniprep kit (Sigma-Aldrich, Dublin, Ireland) was used to isolate recombinant plasmids from *E. coli* Top10 cells according to the manufacturer's instructions. The integrity of the clone was verified by sequencing (Genewiz, Germany). The plasmid pOLEH with His tag was transformed into chemically competent *E. coli* BL21(DE3) according to Sambrook and Russell (34).

**Expression and purification of recombinant *L. delbrueckii* subsp. *bulgaricus* LBP UFSC 2230 OleH putative protein.** For the enzymatic assays, the recombinant putative OleH protein was produced in *E. coli* BL21(DE3) as an N-terminal 6×His tag fusion. LB broth (100 ml) containing kanamycin (50 $\mu$g/ml) was inoculated with a freshly grown overnight culture of *E. coli* BL21(DE3), hosting the oleate hydratase expression plasmid. After culturing at 37°C until the OD at 600 nm ($OD_{600}$) reached 0.6, cultures were induced with 0.5 mM isopropyl-$\beta$-D-1-thiogalactopyranoside (IPTG) at 37°C. After 4 h induction time, cells were harvested by centrifugation (10 min at 6,000 × $g$).

For protein extraction, cells were washed twice with 20 ml wash buffer (50 mM Tris-HCl, 200 mM NaCl, pH 7.0). Lysates were obtained by suspending the pellet in 16 ml lysis buffer (50 mM Tris-HCl, 200 mM NaCl, 0.1 mg/ml lysozyme, and 0.1 mg/ml DNase). The suspension was then sonicated for 3 × 30-s bursts at 16 $\mu$m amplitude (Soniprep 150 sonicator; MSE, Ltd., UK) and harvested by centrifugation. After centrifuging at 6,000 × $g$ at 4°C for 10 min, the supernatant was filtered through a 0.45-$\mu$m membrane filter. Recombinant His-tagged OleH protein was purified by affinity chromatography using fast protein liquid chromatography (FPLC; AKTA purifier; GE Healthcare, Uppsala, Sweden) and loaded onto a HisTrap FF column (GE Healthcare, Uppsala, Sweden). A linear gradient of elution buffer (His buffer kit; 500 mM imidazole; GE Healthcare, Uppsala, Sweden) was applied for 25 column volumes.

Peak fractions containing the desired protein were pooled and analyzed by sodium dodecyl sulfate-polyacrylamide gel electrophoresis (SDS-PAGE). A 10% TruPAGE precast SDS gel (Sigma-Aldrich, Dublin, Ireland) was used for protein separation, and samples were stained with Coomassie brilliant blue G250 dye (Serva Electrophoresis GmbH, Heidelberg, West Germany). The active fractions were collected and dialyzed against 50 mM potassium phosphate buffer (pH 6.0). The dialysis step was carried out in a cold room at 4°C using the Pur-A-Lyzer mega-dialysis kit (Sigma-Aldrich, Dublin, Ireland) system. After dialysis, the solution was used as the purified enzyme. Protein concentration was determined by the Bradford assay kit (Sigma-Aldrich, Dublin, Ireland) procedure according to the manufacturer's instructions.

**Immunoblotting.** The purified protein was subjected to SDS-PAGE, followed by protein transfer to a nitrocellulose membrane at 25 V for 2 h. The membrane was washed with TBST buffer (1× Tris-borate-EDTA [TBE], 0.1% Tween 20) and blocked for 1 h in TBST buffer containing 5% (wt/vol) nonfat dried milk. His-tagged proteins were detected using an anti-His tag antibody (Sigma-Aldrich, St. Louis, USA) in combination with a secondary antibody, both diluted in bovine serum albumin (BSA) buffer (1× TBE, 0.1% Tween 20, and 5% BSA). The membrane was washed three times with TBST buffer and twice with TBS buffer (1× TBE). Immunoreactive bands were detected by chemiluminescence using the Pierce ECL Western blotting substrate (Thermo Scientific, Rockford, USA) according to the manufacturer's instructions.

**Protein sequencing.** Mass spectrometry (MS)-grade Pierce trypsin protease (Thermo Scientific, Rockford, USA) was dissolved in 50 mM acetic acid at a final concentration of 10 ng/$\mu$l. A digestion reaction was carried out for 16 h at 37°C and consisted of 3 $\mu$l of trypsin solution, 10 $\mu$l of OleH, followed by addition of 10 $\mu$l 0.5 mM ammonium bicarbonate. MS was performed on the digested OleH with an Axima TOF2 MALDI-TOF MS (Shimadzu Biotech, Manchester, UK). A 0.5-$\mu$l aliquot of matrix solution (alpha-cyano-4-hydroxycinnamic acid, 10 mg/ml in 50% acetonitrile-0.1% [vol/vol] trifluoroacetic acid) was deposited onto the target and left for 5 s before being removed. The residual solution was allowed to air dry, and 0.5 $\mu$l of the sample solution was deposited onto the precoated sample spot. A 0.5-$\mu$l

aliquot of matrix solution was added to the deposited sample and allowed to air dry. The sample was subsequently analyzed in positive-ion reflectron mode. Protein identification was carried out via peptide mass fingerprinting (PMF) using the Mascot search engine (http://www.matrix-science.com). The monoisotopic positive-ion data ± 0.25 Da were searched using the following parameters: NCBInr database or Swiss-Prot, taxonomy all entries, and trypsin digest with one missed cleavage.

**Molecular modeling and docking analysis.** The three-dimensional model of OleH from *L. delbrueckii* subsp. *bulgaricus* LBP UFSC 2230 was obtained by homology modeling using a Swiss-Model server (35), based on the crystallographic structure of the hydratase from *Lactobacillus acidophilus* bound to LA (PDB ID 4IA6; resolution, 1.80 Å), with a sequence identity of 72.3%. The parameters QMEAN4 (maximum, −3.5), GMQE (values closer to 1), coverage greater than 95%, and values of $C\beta$, solvation, and torsion (most positive or close to zero) were also considered. PROCHECK (36) was used to verify the reliability of the three-dimensional models. The program CASTp (37) was used to calculate the cavities sites in the three-dimensional model with a radius probe of 1.3 Å. Tunnels and access were mapped using Caver 3.0 software (38).

An *in silico* docking experiment was carried out in the AutoDock Vina program (39). Docking boxes were constructed in the Chimera software (40) based on the cavities predictions and the pockets assigned by Volkov et al. (23). The tested ligands were obtained from Zinc database (41), LA (ZINC4474613), net charge −1; ricinoleic acid (12-hydroxy-9-*cis*-octadecenoic acid [RA]) (ZINC3875925), net charge, −1; and flavin adenine dinucleotide (FAD) (ZINC8215434), net charge, −3. The most stable docking position (for each ligand) was selected according to its position in the binding pocket, the lowest energy, full fitness, and $\Delta G$ in Chimera (version 1.12.2). The analyzed docking positions were selected considering binding affinity, $\Delta G$, and position in the binding pocket, using the PyMOL software (42). The interactions between residues and the ligands were identified using the LigPlot software (43). Docking positions were compared to the corresponding ligand location in the experimental structure and had the root mean square deviation (RMSD) of docked poses calculated by DockRMSD server (44) and Chimera program (40).

**Effects of temperature and pH on enzyme stability.** The effect of pH on the stability of the putative OleH from *L. delbrueckii* subsp. *bulgaricus* LBP UFSC 2230 was evaluated by buffer exchange, dialyzing the purified protein against 50 mM potassium phosphate buffer, pH 6.0; 0.1 M Tris-HCl, 150 mM NaCl buffer, pH 7.5; or 20 mM sodium phosphate, 0.5 M NaCl buffer, pH 8.0. To investigate the effect of temperature on the stability of the OleH from *L. delbrueckii* subsp. *bulgaricus* LBP UFSC 2230, the enzyme was incubated at different temperatures ranging from 25 to 37°C.

**Oleate hydratase activity assay.** The ability of isolates to convert LA (Sigma-Aldrich, Dublin, Ireland) and RA (Sigma-Aldrich, Dublin, Ireland) to CLA was performed as follows. Enzymatic reactions were performed in triplicate. The standard reaction conditions consisted of 1 ml of reaction mixture (50 mM potassium phosphate buffer, pH 6.0) containing 0.75 mg/ml purified OleH, 20 $\mu$M FAD, 10 mM individual unsaturated fatty acid, and 2% (vol/vol) ethanol. The reactions were carried out for 15, 30, 45, and 60 min anaerobically at 37°C. Two negative controls were used consisting of all the components except substrates and all the components except the purified protein.

**Biotransformation of LA and RA.** Whole-cell biocatalysis was performed based on Fibinger et al. (45), with modifications. The recombinant *E. coli* BL21(DE3) was cultivated in LB medium with kanamycin (50 $\mu$g/ml) at 37°C with shaking at 200 rpm. The expression of the target gene was induced as described earlier. Two milliliters of the cell cultures were washed with 50 mM potassium phosphate buffer, pH 6.0 (filter sterilized [0.22 $\mu$m]) and resuspended in 1 ml of the same buffer. Biotransformation reactions were conducted by incubating cell suspensions with 10 mM LA or RA emulsified with 2% (wt/vol) of Tween 80 (Sigma-Aldrich, Dublin, Ireland) at 37°C under anaerobic conditions.

**Lipid analysis by gas chromatography.** Before lipid extraction, $C_{13:0}$ tridecanoic acid (99% pure; Sigma-Aldrich, Dublin, Ireland) was added to the reaction mixture as an internal standard. Lipids were extracted from 1 ml of the reaction mixtures using hexane/isopropanol according to Coakley et al. (5) and then concentrated under a stream of nitrogen at 45°C. The resulting lipids were methylated as described by Stanton et al. (46) with 4% methanolic HCl in methanol (vol/vol) at 50°C for 20 min. After the addition of 2 ml water saturated with hexane, the resulting fatty acid methyl esters (FAME) were extracted with 5 ml hexane.

FAME analysis was performed by gas chromatography (GC) on an Agilent 7890B GC, equipped with a GC80 autosampler (Agilent Technologies, Little Island, Cork, Ireland) and a flame ionization detector. The column was a Select FAME capillary column (100 m by 250 $\mu$m i.d., 0.25 $\mu$m phase thickness; part number CP7420; Agilent Technologies). The injector was held at 250°C for the entire run and was operated in split mode using a split ratio of 1:10, and the injection volume was 1 $\mu$l. The inlet liner was a split gooseneck liner (part no. 8004-0164; Agilent Technologies). The column oven was held at 80°C for 8 min and raised to 200°C at 8.5°C/min and held for 55 min. The total run time was 77.12 min. The flame ionization detector was operated at 300°C. The carrier gas was helium and was held at a constant flow of 1.0 ml/min.

Results were processed using OpenLab CDS ChemStation Edition software version Rev.C.01.05 (Agilent Technologies). Standard curves for FAME analysis along with in-run quality control samples were prepared using an Agilent 7696A Sample Prep Workbench instrument (Agilent Technologies). The standard mix of CLA *cis*-9, *trans*-11-$C_{18:2}$, and *trans*-10, *cis*-12-$C_{18:2}$ (catalog no. UC-59M) were purchased from Nu-Chek Prep Inc. (Elysian, MN). FAME triglyceride standard mix containing C18:0 to C20:0 methyl esters (cat. no. 18916-1 AMP) and internal standard tridecanoic acid (catalog no. T0502) were purchased from Sigma-Aldrich (Dublin, Ireland).

## ACKNOWLEDGMENTS

This research was supported by the Coordenação de Aperfeiçoamento Pessoal de Nível Superior (CAPES), financial code 001, and APC Microbiome Ireland, a research center funded by Science Foundation Ireland (SFI), through the Irish Government's National Development Plan.

We are grateful to Lucas Mascarin (Department of Microbiology, Immunology and Parasitology, Federal University of Santa Catarina, Florianópolis, Santa Catarina, Brazil) for contributing with AKTA Protein Purification System, Paula O'Connor (Teagasc Food Reasearch Centre, Moorepark, Fermoy, Co. Cork, Ireland) for contributing with protein sequencing, Miguel Ullivarri (APC Microbiome Ireland, Biosciences Institute, University College Cork, College Road, Cork, Ireland) and Taís Kuniyoshi (Biochemical and Pharmaceutical Technology Department, Faculty of Pharmaceutical Sciences, University of São Paulo) for contributing with the cloning protocols, and Conall Strain (Teagasc Food Reasearch Centre, Moorepark, Fermoy, Co. Cork, Ireland) and Brian Healy (Teagasc Food Reasearch Centre, Moorepark, Fermoy, Co. Cork, Ireland) for useful comments on the manuscript.

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
