## [Reviewer comments · Microbiology Spectrum]

**Microbiology
Spectrum**

Oleate Hydratase in *Lactobacillus delbrueckii* subsp. *bulgaricus* LBP UFSC 2230 Catalyzes the Reversible Conversion Between Linoleic Acid and Ricinoleic Acid

Gabriela Kuhl, Ricardo Mazzon, Brenda Simas Porto, Tâmela Zamboni Madaloz, Guilherme Razzera, Daniel Patricio, Kevin Linehan, Grace Ahern, Harsh Mathur, R. Paul Ross, Catherine Stanton, and Juliano De Dea Lindner

Corresponding Author(s): Juliano De Dea Lindner, Federal University of Santa Catarina (UFSC)

Review Timeline:

Submission Date:

August 31, 2021

Accepted:

September 6, 2021

Editor: Jeffrey Gralnick

Reviewer(s): The reviewers have opted to remain anonymous.

Transaction Report:

DOI: <https://doi.org/10.1128/Spectrum.01179-21>

September 6, 2021

Prof. Juliano De Dea Lindner
Federal University of Santa Catarina (UFSC)
Food Science and Technology
UFSC - CCA - CAL
Florianópolis, SC 88034-001
Brazil

Re: Spectrum01179-21 (Oleate Hydratase in *Lactobacillus delbrueckii* subsp. *bulgaricus* LBP UFSC 2230 Catalyzes the Reversible Conversion Between Linoleic Acid and Ricinoleic Acid)

Dear Prof. Juliano De Dea Lindner:

Based on your responses and revisions to the prior round of review, your manuscript has been accepted, and I am forwarding it to the ASM Journals Department for publication. You will be notified when your proofs are ready to be viewed.

Sincerely,

Jeffrey Gralnick
Editor, Microbiology Spectrum
